# Mechanochemistry as an Alternative Method of Green Synthesis of Silver Nanoparticles with Antibacterial Activity: A Comparative Study

**DOI:** 10.3390/nano11051139

**Published:** 2021-04-28

**Authors:** Matej Baláž, Zdenka Bedlovičová, Nina Daneu, Patrik Siksa, Libor Sokoli, Ľudmila Tkáčiková, Aneta Salayová, Róbert Džunda, Mária Kováčová, Radovan Bureš, Zdenka Lukáčová Bujňáková

**Affiliations:** 1Department of Mechanochemistry, Institute of Geotechnics, Slovak Academy of Sciences, Watsonova 45, 04001 Košice, Slovakia; kovacovam@saske.sk (M.K.); bujnakova@saske.sk (Z.L.B.); 2Department of Chemistry, Biochemistry and Biophysics, University of Veterinary Medicine and Pharmacy, Komenského 73, 04181 Košice, Slovakia; zdenka.bedlovicova@uvlf.sk (Z.B.); patrik.siksa@student.uvlf.sk (P.S.); libor.sokoli@uvlf.sk (L.S.); aneta.salayova@uvlf.sk (A.S.); 3Advanced Materials Department, Jožef Stefan Institute, Jamova cesta 39, 1000 Ljubljana, Slovenia; nina.daneu@ijs.si; 4Department of Pharmacology and Toxicology, University of Veterinary Medicine and Pharmacy, Komenského 73, 04181 Košice, Slovakia; 5Department of Microbiology and Immunology, University of Veterinary Medicine and Pharmacy, Komenského 73, 04181 Košice, Slovakia; ludmila.tkacikova@uvlf.sk; 6Institute of Materials Research, Slovak Academy of Sciences, 04001 Košice, Slovakia; rdzunda@saske.sk (R.D.); rbures@saske.sk (R.B.)

**Keywords:** mechanochemistry, Ag nanoparticles, green synthesis, solid-state synthesis, antibacterial activity

## Abstract

This study shows mechanochemical synthesis as an alternative method to the traditional green synthesis of silver nanoparticles in a comparative manner by comparing the products obtained using both methodologies and different characterization methods. As a silver precursor, the most commonly used silver nitrate was applied and the easily accessible lavender (*Lavandula angustofolia* L.) plant was used as a reducing agent. Both syntheses were performed using 7 different lavender:AgNO_3_ mass ratios. The synthesis time was limited to 8 and 15 min in the case of green and mechanochemical synthesis, respectively, although a significant amount of unreacted silver nitrate was detected in both crude reaction mixtures at low lavender:AgNO_3_ ratios. This finding is of particular interest mainly for green synthesis, as the potential presence of silver nitrate in the produced nanosuspension is often overlooked. Unreacted AgNO_3_ has been removed from the mechanochemically synthesized samples by washing. The nanocrystalline character of the products has been confirmed by both X-ray diffraction (Rietveld refinement) and transmission electron microscopy. The latter has shown bimodal size distribution with larger particles in tens of nanometers and the smaller ones below 10 nm in size. In the case of green synthesis, the used lavender:AgNO_3_ ratio was found to have a decisive role on the crystallite size. Silver chloride has been detected as a side-product, mainly at high lavender:AgNO_3_ ratios. Both products have shown a strong antibacterial activity, being higher in the case of green synthesis, but this can be ascribed to the presence of unreacted AgNO_3_. Thus, one-step mechanochemical synthesis (without the need to prepare extract and performing the synthesis as separate steps) can be applied as a sustainable alternative to the traditional green synthesis of Ag nanoparticles using plants.

## 1. Introduction

Green synthesis of silver nanoparticles (Ag NPs) with antibacterial activity is a well-established environmentally friendly approach [1,2]. In contrast with traditional synthetic approaches, it does not require the use of toxic chemicals or solvents, as the synthesis is performed using various biological organisms, plants being the most commonly used [3,4,5]. Nature provides us with an almost infinite number of plants, out of which a huge number has already been used for the production of Ag NPs. The plants differ in their ability to reduce Ag^+^ ions into Ag^0^, which is a key process during the synthesis.

In the vast majority of the papers on green synthesis, the authors do not investigate the actual content of residual silver nitrate (most common Ag precursor) in the produced nanosuspension, so they do not have the information about the actual reaction progress. Although there is an example of AgNO_3_ quantification in [6], usually it is not done and the termination of the reaction is determined based just on the saturation of absorbance in the ultraviolet–visible (UV-Vis) spectra, which does not have to necessarily take place due to consumption of all introduced AgNO_3_, but might be also a result of exhaustion of available reduction components from the plant. In a number of publications, the produced Ag nanoparticles are purified from non-reacted AgNO_3_ by centrifugation (e.g., in [7,8,9,10]) and the properties (including e.g., antibacterial activity) are investigated afterwards. This is the correct approach and the exact concentration of Ag NPs can be then quantified. However, in the majority of the papers, either the as-received nanosuspension containing also non-reacted AgNO_3_ is used for further experiments (and the results can of course be significantly affected then), or centrifugation is applied just for the sake of obtaining representative results from powder characterization (e.g., X-ray diffraction to have the proof of Ag^0^ presence). In our manuscript, we have obtained the proof of reaction incompletion and we are stressing out the necessity of the investigation of the reaction progress.

Within the rich plethora of the plants that have been applied for the green synthesis of Ag nanoparticles so far, also lavender (*Lavandula angustofilia* L.), as a representative of common plants has been successfully used [11,12,13,14]. Ag NPs prepared by using this plant have been already applied for 4-nitrophenol degradation and H_2_O_2_ sensing [13], antioxidant activity [12] and biological activity (including antibacterial) [14].

Mechanochemical synthesis has emerged as an alternative method to traditional chemistry with very broad application potential [15,16,17,18,19,20,21,22,23,24,25,26,27] and has been recently listed among 10 chemical innovations that will change our world [28]. It has been successfully applied for nanoparticles production (including Ag) in the past [17]. However, there are just few reports on the application of mechanical force to prepare Ag nanoparticles using natural materials [29,30,31,32,33,34,35,36]. There has been no publication directly comparing the bio-mechanochemical and green synthesis so far. However, a group of Iranian researchers published a study in which they clearly compared green synthesis and solid-state synthesis by hammer milling [33]. The latter process was followed by a high-temperature treatment, thus neglecting the environmentally friendly character of their approach.

In this paper, we clearly show the pros and cons of both approaches in relation to the concentration of AgNO_3_ precursor on a selected example. The aim was to show mechanochemistry as a competitive and alternative method to well-established green synthesis of Ag nanoparticles using plant extracts. The combination of mechanochemistry and green synthesis (also entitled bio-mechanochemical synthesis) for Ag nanoparticles production brings about two environmentally friendly aspects: (i) the use of natural material as reducing agent and (ii) the possibility to perform the synthesis in the solid-state in one step (without the need of a separate step of extract preparation). Moreover, by using our approach, the antibacterial activity can be ascribed just to the prepared Ag NPs, as the unreacted AgNO_3_ has been effectively washed out, by contrast with the classical green synthetic approach. To the best of our knowledge, such a comparative study has not been reported until now.

## 2. Materials and Methods

### 2.1. Materials

Silver nitrate, AgNO_3_ (99.9% purity) was purchased from Mikrochem, Pezinok, Slovakia. Dried leaves and flowers of lavender plant (*Lavandula angustifolia* L.) were purchased from Agrokarpaty Plavnica, Plavnica, Slovakia.

### 2.2. Synthesis of Silver Nanoparticles (Ag NPs)

The synthesis process was investigated for 7 different lavender:AgNO_3_ ratios. The comparison of actual masses of the lavender plant and AgNO_3_ for both synthetic approaches, together with the plant:AgNO_3_, ratio is provided in Table 1 below.

In the following text, the samples prepared by green and mechanochemical synthesis are referred to as LEV-Ag-GS-14.87…0.99 and LEV-Ag-MS-14.87…0.99, respectively. Also the lavender:AgNO_3_ mass ratio is abbreviated as the L:A ratio. An additional experiment with mechanochemical synthesis for 120 min using the L:A ratio 1:1 was performed and this sample has been also included into the table (LEV-Ag-1.00-120).

#### 2.2.1. Green Synthesis

Extract Preparation

The extract was prepared by stirring 600 mg of powdered lavender (*Lavandula angustifolia* L.) leaves in 12 mL of demineralized water for 2 h at room temperature. After filtration the extract was used for silver nanoparticles synthesis.

Silver Nanoparticles Synthesis

Silver nanoparticles were synthesized by adding extract to freshly prepared aqueous silver nitrate solution (2.2; 5.5; 11; 16.5; 22; 27.5 and 33 mM) in the ratio 1:9. The reaction mixtures were heated to 80 °C. The formation of Ag NPs was observed visually by detecting colour changes and by monitoring UV-Vis spectra every minute in region of 350–750 nm. The syntheses were stopped when the absorbance in monitored region did not increase anymore. The final samples after green synthesis are denoted as LEV-Ag-GS-14.87-0.99 (the number represents the lavender: AgNO_3_ mass ratio, for details, please see Table 1). The obtained nanosuspensions were used for further characterization and antibacterial tests. In order to obtain enough material for further characterization, green synthesis was performed on a larger scale (in a round-bottom flask), the reaction mixtures were dried at laboratory temperature and the received powder was subjected to X-ray diffraction analysis.

#### 2.2.2. Mechanochemical Synthesis of Ag NPs

Prior to mechanochemical synthesis, the particle size of the plant material was reduced using a commercial mixer and the fraction with particles smaller than 1 mm used for further experiments.

For the mechanochemical synthesis, different amounts of lavender plant and AgNO_3_ powders were introduced into the milling pot. The overall mass of the milled mixture was always 3 g. Milling was performed in Pulverisette 7 Premium line planetary ball mill (Fritsch, Idar-Oberstein, Germany) under the following conditions: air atmosphere, 15 tungsten carbide milling balls with the diameter 10 mm, ball-to-powder ratio 37, milling speed 500 rpm, milling time 15 min (or 2 h for a selected experiment). After milling, 1 g of the obtained powder was washed with 100 mL distilled water mainly to remove non-reacted AgNO_3_. The washed samples after mechanochemical synthesis are denoted as LEV-Ag-MSW-14.87-0.99 (W stands for washed). The solid residue was used for further characterization and antibacterial tests.

### 2.3. Characterization

The silver nanoparticles formation was monitored by Cary 60 UV-Vis spectrophotometer (Agilent Technologies, Santa Clara, CA, USA) with a water-cooled Peltier thermostatted cell holder for heating.

The particle size distribution in the nano-range was measured by photon cross-correlation spectroscopy using a Nanophox particle size analyzer (Sympatec, Clausthal-Zellerfeld, Germany). A portion of each nanosuspension was diluted with the distilled water to achieve a suitable concentration for the measurement. This analysis was performed using a dispersant refractive index of 1.33. The measurements were repeated 3 times for each sample.

Transmission electron microscopy (TEM) analyses were performed on a 200 kV microscope JEM 2100 (Jeol, Tokyo, Japan). A droplet of the samples prepared by green synthesis was applied onto a lacy carbon-coated copper grid. The samples prepared using the dry mechanochemical synthesis approach were first dispersed in absolute ethanol, sonicated for a few minutes and then applied to the copper grid. The grids were dried and additionally coated with a thin layer of carbon to prevent charging.

The X-ray diffraction (XRD) patterns were obtained using a D8 Advance diffractometer (Bruker, Billerica, MA, USA) with CuKα (40 kV, 40 mA) radiation. All samples were scanned from 15° to 70° with steps 0.05° and 15 s counting time. Pure plant was measured only between 15 and 50° using step size 0.05° and step time 15 s.

The grain size analysis was performed using a particle size laser diffraction analyzer Mastersizer 2000E (Malvern Panalytical, Malvern, UK) in the dry mode. Each sample was measured three times.

The content of silver in solid samples was analyzed using an atomic absorption spectrometer SPECTRAA L40/FS (Varian, Crawley, UK). Prior to the measurement, the samples were dried at 50–60 °C for 1 h and subsequently 0.1 g was dissolved in 10 mL of concentrated nitric acid. The solution was then diluted with distilled water to get the volume of 100 mL.

The morphology and size of the products was investigated by a scanning electron microscope Tescan Vega 3 LMU (TESCAN, Brno, Czech Republic) using accelerating voltage 20 kV. In order for the samples to be conductive, the powder was covered by a layer of gold on a FINE COAT ION SPUTTER JFC-1100 fy (JEOL, Akishima, Japan). To obtain the information about chemical composition, the energy-dispersive X-ray spectroscopy (EDS) analyzer Tescan: Bruker XFlash Detector 410-M (TESCAN, Brno, Czech Republic) was used. The same device was used to record elemental maps.

### 2.4. Antibacterial Activity

The antibacterial properties of the samples were evaluated by the agar well diffusion method by slight modification of the process reported in [37]. The tested bacteria (*S. aureus* CCM 4223 and *E. coli* CCM 3988) were obtained from the Czech collection of microorganisms (CCM, Brno, Czech republic). The procedure was as follows:Bacteria were cultured overnight, aerobically at 37 °C in LB medium (Sigma-Aldrich, Saint-Louis, MO, USA) with agitation. After this, bacteria were mix with 50% glycerol (Mikrochem, Pezinok, Slovakia) and frozen glycerol stock cultures were maintained at −20 °C. Before the experimental use, cultures were transferred to LB medium and incubated for 24 h and used as the source of inoculum for each experiment.Plate count agar (HIMEDIA, Mumbai, India) medium was cooled to 42 °C after autoclaving, inoculated overnight with liquid bacterial culture to a cell density of 5 × 10^5^ colony-forming units cfu/mL.20 mL of this inoculated agar was pipetted into a 90 mm diameter Petri dish.Once the agar was solidified, five mm diameter wells were punched in the agar and filled with 50 µL of samples prepared in the form of suspensions (either directly the nanosuspensions prepared by green synthesis or 50 µL of the suspensions prepared by dispersing 20 mg of LEV-Ag-MS samples in 1 mL of distilled water). Gentamicin sulfate (Biosera, Nuaille, France) with the concentration 30 mM was used as a positive control.The plates were incubated for 24 h at 37 °C.Afterwards, the plates were photographed and the inhibition zones were measured by the ImageJ 1.53e software (U. S. National Institutes of Health, Bethesda, MD, USA). The values used for the calculation are mean values calculated from 3 replicate tests.

The antibacterial activity was calculated by applying the formula reported in [37]:%RIZD = [(IZD sample − IZD negative control)/IZD gentamicin] × 100
where RIZD is the relative inhibition zone diameter (%) and IZD is the inhibition zone diameter (mm). As a negative control, the inhibition zones of distilled water equal to 0 were taken. The inhibition zone diameter (IZD) was obtained by measuring the diameter of transparent zone and subtracting the size of the wells (5 mm).

## 3. Results and Discussion

### 3.1. Justification of Experimental Setup

The idea of this work was to compare two approaches of Ag NPs preparation using the lavender plant, namely the established liquid-phase green synthesis with the novel mechanochemical one. In order for the experiments to be comparable, the ratios of silver nitrate and plant material have to be same. To achieve this, the following hypothesis described below was used.

As already mentioned in the experimental part, 12 mL of water extract was prepared by using 600 mg of lavender plant as a first step of the green synthesis. Subsequently, 0.3 mL of the extract was used. This accounts for 15 mg of plant material. This amount was fixed during all green synthesis experiments. We were changing the amount of AgNO_3_ introduced into the system. For example in 2.7 mL AgNO_3_ solution with the lowest concentration (2.2 mM), the actual mass of AgNO_3_ in the experiment is 1.0090 mg. If we divide the actual mass of lavender plant by the actual mass of AgNO_3_, we obtain the lavender:AgNO_3_ mass ratio. In the case of mechanochemical synthesis, we only used these ratios and re-calculated it so the whole mixture would be 3 g.

We are of course aware that such an assumption is not completely correct and can be taken just as approximate. For example, just water-soluble species were actually present in the extract during the green synthesis. That is, after the evaporation of water from the extract, the obtained solid weighed only 5.2 mg representing 34.5% out of 15 mg of lavender plant introduced into the extraction process. However, these water-soluble compounds are mainly responsible for the desired reduction process. Thus, the powder of water-soluble plant compounds would be by far more reactive than the whole plant used during the mechanochemical synthesis, where many plant species stay intact. It also has to be taken into account that the mechanical force increases reactivity so the compounds which did not dissolve in water during extraction, can take place in the solid-state reaction during the mechanochemical synthesis. The same methodology by providing the same amount of dry plant (regardless of the fact whether or not the extract was prepared) with respect to AgNO_3_ was used in [33].

### 3.2. Green Synthesis

#### Ultraviolet-Visible (UV-Vis) Measurements

The primary detection of bio-reduction of Ag^+^ ions to colloidal Ag NPs can be observed by visual colour change and is usually analysed by UV-Vis spectral analysis. The green synthesis of Ag NPs was realized in situ directly in a cuvette used for UV-Vis measurements by heating the reaction mixture to 80 °C using Peltier heating equipment. The light absorption pattern of the mixture of lavender extracts and silver nitrate with the corresponding concentration was monitored in the range of 350–750 nm. The reaction progress was then monitored by measuring the UV-Vis spectra every minute. However, for the L:A ratios 1.98 and lower, the reaction was finished until 1 min (the absorbance measured after two minutes was not higher than after the first minute). The time-resolved UV-Vis spectra for the three lavender-richest samples, together with the plot showing the dependence of the absorbance maximum on time for all LEV-Ag-GS samples can be found in the Appendix A.

The formation of silver nanoparticles can be monitored by increasing intensity of the peak usually located at wavelengths in the range of 400–470 nm corresponding to the surface plasmon resonance (SPR) of Ag NPs. A significant increase of the intensity of the peak located exactly in this range with reaction time can be well seen in all LEV-Ag-GS-14.87–2.97 samples. The maximum absorbance for the sample with L:A ratio 14.87 was achieved in 8 min (Appendix A) and with decreasing L:A ratio, the necessary time decreased (namely to 3 and 2 min for LEV-Ag-GS-5.95 and 2.97, respectively, see Appendix A). Such a phenomenon is in accordance with other reports [38,39]. In the case of other LEV-Ag-GS samples with lower L:A ratios, the absorbance maximum is reached within the 1st minute (Appendix A). Although the absorbance increases more rapidly with decreasing L:A ratio, it does not mean that the reaction will be completed faster. As more AgNO_3_ is introduced, more Ag NPs can be potentially formed, thus leading to a more intensive brown color, thus reaching the limit of the appropriateness of UV-Vis spectroscopy (for the Lambert-Beer law to be valid, the absorbance values should be lower than 1). In ref. [6], the extract:AgNO_3_ 1mM solution ratio used was 1:10 and the synthesis was performed at 60–65 °C. The red colour documenting the successful synthesis was observed after 3 h, but the increase in absorbance was observed until 96 h. In ref. [12], the ultrasound-assisted synthesis performed at 30 °C by mixing 20 mL 1mM AgNO_3_ solution with the same amount of lavender flowers extract was terminated after 10 min.

In general, very intensive peaks located in the range of 413–474 nm were observed for LEV-Ag-GS samples. The colours of prepared nanosuspensions were red. In addition to being dependent on the precursor concentration, the reaction time of Ag^+^ ions reduction is also temperature-dependent, as increasing the reaction temperature increases the reduction rate and finally shortens the reaction time necessary for Ag NPs synthesis [40]. Because of this, we have decided to use higher temperature (80 °C) and the bio-reduction took only few minutes (1–8 min) depending on the concentration of AgNO_3_.

A plot summarizing all final UV-Vis spectra for LEV-Ag-GS samples obtained at different L:A ratios is presented in Figure 1.

Broad absorption spectra of colloid silver nanoparticles can be observed for all samples (Figure 1a), however, the position of the maximum of SPR peak is shifting with L:A ratio (Figure 1b). In the case of highest AgNO_3_ concentration (L:A ratio 0.99) the SPR peak with maximum at 514 nm was observed. The SPR peak maximum is blue-shifted in an exponential manner (Figure 1b) with increasing L:A ratio to reach 422 nm in for the lavender-richest sample with L:A ratio 14.87 (the wavelength of the maximum is the same also for the samples with L:A ratio 5.95 and 2.97). The maximum is shifted (displaced) depending on the concentration of silver nitrate used as a precursor. The surface plasmon resonance of the nanoparticles formed when using higher precursor concentrations will be red-shifted to the higher wavelengths as larger particles are formed [41,42,43]. For example in ref. [13], Ag NPs prepared by lavender extract exhibited the SPR band at the wavelength of 440 nm. The maximum was shifted with reaction time to 460 nm which was ascribed to the increase in crystallite size. In our case, we have a similar situation, but the blue shift in our case is not time-dependent (as the reactions were very fast), but concentration-dependent. We have observed a similar situation when using *Origanum vulgare* L. as a reducing agent in the past [44].

### 3.3. Mechanochemical Synthesis

The mechanochemical synthesis is known to be an interesting alternative to the classical preparation of nanocrystalline materials, including Ag nanoparticles. The utilization of mechanochemical protocols for the synthesis of Ag nanoparticles using natural materials is a rather unexplored field. As described in the experimental section, a set of samples with different L:A ratios in the form of powders was prepared in this work.

Although the experimental setup with regards to L:A ratio was clear, we were looking for optimal milling conditions. In our previous study, we used 2 h of treatment at revolutions of 500 min^−1^ [32]. In that work, the plant:AgNO_3_ ratio was set to 1. Thus, we performed a comparative experiment using equal amounts of AgNO_3_ and lavender for two hours of milling (sample is entitled as LEV-Ag-MS-1.00-120 min). However, for the mechanochemical synthesis to be competitive with the green synthesis running usually for tens of minutes (at higher temperatures), we tried to reduce the milling time to 15 min. As the highest ratio we used in this study for the investigation of different L:A ratios (see Table 1) was almost the same (0.99), we considered this experiment to provide the same output as the one performed at L:A ratio 1.00. The XRD patterns of the as-received powders milled for 15 and 120 min can be found in the ESI (Appendix A).

As can be seen, the content of AgNO_3_ is very high in both samples, by contrast with [32], where its diffractions were not observed already after 45 min of milling. Thus, lavender seems to be much weaker reducing agent than *Origanum vulgare* L. In the present case, the diffractions of Ag^0^ seem to have increased as a result of prolonged treatment, thus the mechanochemical approach has been partly effective in promoting the reaction progress. Prolonged milling up to two hours was capable of shifting the reaction progress a bit, as according to the Rietveld refinement, the content of Ag^0^ for LEV-Ag-MS-0.99 (treated for 15 min) and LEV-Ag-MS-1.00-120 (treated for two hours) was 3% and 9%, respectively. However, we wanted to propose an alternative method to the green synthesis, therefore it was not appropriate to mill the mixtures for such a long time as 2 h and we performed the MS experiments at different L:A mass ratios for 15 min, despite lower yield of Ag^0^ at the highest used AgNO_3_ amount. As AgNO_3_ amounts for all the other experiments were lower, it was hypothesized that the reaction conversion should improve with increasing L:A ratios.

We have tried to record the UV-Vis spectra of the products obtained by milling to directly compare them with those obtained for the green synthesis (see Figure 1), however, the absorbance maximum which should correspond to surface plasmon resonance of Ag nanoparticles was not registered in either of the samples. This was surprising, as in our previous studies, we succeeded in obtaining UV-Vis spectra clearly showing the presence of Ag NPs [32,45]. Because of failing to obtain representative UV-Vis spectra, we turned to X-ray diffraction to gain proof of the presence of nanocrystalline elemental silver formation.

#### 3.3.1. Phase Composition and Crystal Structure of the Washed Powders

The progress of the mechanochemical synthesis was monitored by means of X-ray diffraction. For the sake of direct comparison of two synthetic approaches, the XRD patterns of as-received powders with four lowest L:A mass ratios are reported later (in Section 3.4.1). At higher L:A ratios, the amorphous halo coming from the plant makes it difficult to obtain any diffraction peaks. In our recent study, we have discovered that at high plant:AgNO_3_ ratios, a specific role can be played by chlorine coming from the plant matrix [45]. It seems that if there is any chlorine available for reaction, it will react preferentially with AgNO_3_ to form AgCl and only after the exhaustion of available chloride ions, the reduction of Ag^+^ to Ag^0^ by the reducing groups of the plant matrix starts. This has been specifically shown for the plants *Thymus serpyllum* L. and *Thymus vulgaris* L., where at plant:AgNO_3_ mass ratios 100, the diffractions corresponding to AgCl were more pronounced than those of the Ag^0^ product.

Chlorine is most probably present also in the plant matrix of lavender. In some of the studies reporting the green synthesis of Ag NPs using the lavender plant, the presence of AgCl has been also detected. For example in [46], the authors did not observe the reflections of Ag^0^, just the ones belonging to AgCl (although in this case, another type of lavender, namely *Lavandula dentata* L. leaves have been used). In our recent paper where different plants, including *Lavangula angustofolia* L. have been used [47], AgCl has been detected by selected area diffraction (SAED) used with transmission electron microscopy (TEM). In other studies, no mention about AgCl is provided [11,12,13].

As will be shown later, no AgCl could be detected in the four Ag-richest as-received samples in the present study, however, the L:A ratios are very low (from 1 to 5.95) in comparison with 50 and 100 used in [45]. Moreover, as will be shown, plenty of non-reacted AgNO_3_ was detected which further complicates potential detection of AgCl.

In our previous studies, we found out that washing the as-received powder with water is necessary for the elimination of the backward transformation of Ag^0^ to AgNO_3_ [48]. Because of this, we subjected the as-milled powders obtained at all L:A ratios to washing. The XRD patterns of the washed powders (LEV-Ag-MSW samples) can be found in Figure 2.

The Bragg peaks corresponding to elemental silver can be well seen for the samples LEV-Ag-MSW-0.99-5.95. Ag^0^ peaks exhibited even higher intensity for the sample treated for 120 min (LEV-Ag-MSW-1.00-120). In this case, also a small shift of Ag^0^ diffraction peaks was observed, which means a change in the lattice parameters of its face-centered cubic unit cell. In addition to Ag^0^, also silver chloride AgCl was identified (its main peak is located at around 32°) in all samples (although its content for the sample treated for 2 h seems to be very low). Its formation during both green and mechanochemical syntheses was observed also when using other plants [31,45,49,50,51,52]. Thus, after the removal of residual silver nitrate and some water-soluble lavender compounds, the presence of limited amount of AgCl became observable. We are of the opinion that AgCl has been formed already during mechanochemical synthesis, however, the small reaction progress hampered the identification of AgCl in the as-received samples. For the samples with higher L:A ratios (LEV-Ag-MSW-5.95 and 14.87), the peaks of elemental silver are very broad and of low intensity, meaning the produced NPs are very small and they are most probably scarcely distributed in the plant matrix. By contrast, the peaks of AgCl can still be identified well and their intensity is higher than that of Ag^0^, which is in accordance with the results detected for other plants at high plant:AgNO_3_ ratios reported earlier [45] and with the hypothesis outlined therein about the preferential formation of AgCl. No residual AgNO_3_ was evidenced in neither of the washed samples, thus the washing process was successful.

Rietveld refinement of the XRD patterns presented in Figure 2 was performed in order to investigate the effect of L:A ratio on the crystallite size of the produced Ag and AgCl. The results are summarized in Figure 3. Because of a very high amount of amorphous plant material and very low content of Ag^0^ in LEV-Ag-14.87-MSW sample, it was not possible to perform representative Rietveld refinement in this case.

In general, the increasing L:A ratio led to a decrease of crystallite size of both AgCl and Ag (Figure 3). The prevalence of the plant material led to an effective stabilization of smaller nanoparticles, whereas the larger concentrations of AgNO_3_ led to the merging of smaller non-stabilized particles into larger ones. This is in accordance with [45]. Interestingly, the crystallite size of AgCl seems to increase with the growing amount of plant until L:A ratio 1.48 and then a decrease is observed. For crystallite size of Ag^0^, a decrease was observed with increasing lavender:AgNO_3_ ratio, with a small exception for the peculiar sample with L:A ratio 1.98, where the crystallite size seems to be too small and jumps out of the general trend. Longer milling time did not influence significantly the crystallite size of the produced Ag^0^, however, that of AgCl seems to have decreased (the size around 13 nm was detected). In general, the calculated crystallite size for mechanochemically synthesized Ag nanoparticles using the L:A 0.99 ratio were smaller than those reported when using the same plant:AgNO_3_ ratio for other plant-mediated mechanochemically synthesized Ag NPs (for *Origanum vulgare* L., *Thymus serpyllum* L., *Thymus vulgaris* L. and *Sambucus nigra* L.-mediated syntheses, the values 34, 27, 22 and 18 nm, respectively, have been reported [32,45]). In the reports on green synthesis using lavender, the crystallite sizes in the range 10–80 nm [13] and around 20 nm as determined from scanning electron microscopy (SEM) [12] were obtained.

#### 3.3.2. Atomic Absorption Spectrometry

In order to find out how much silver was effectively stabilized during milling, we analyzed its amount in the solid samples after the mechanochemical synthesis (MS samples), i.e. after milling (prior to washing, amount of silver denoted as M) and after washing in the MSW samples (amount of silver denoted as W) using atomic absorption spectroscopy. We also calculated the ratio between of the values detected x after and before washing (W/M). The higher the W/M ratio, the better the stabilization. The results are provided in Table 2 below.

The washing process led to a significant weight loss. This can be attributed to washing out of Ag species (both non-reacted AgNO_3_ and non-stabilized Ag nanoparticles) and water-soluble compounds from the lavender plant. A gradual decrease in the mass loss with the increasing content of AgNO_3_ introduced into the milled mixture can be observed.

The comparison of theoretical Ag content and the one detected after milling tells us that there was a small loss of Ag during milling, as in all samples (with one exception), the values after milling are lower by 0.5–4%. It seems that some amount of silver-containing compound (it can be either silver nitrate or the produced elemental silver) could cover the milling media (balls, or chamber walls) and after the extraction of the sample, remains in the milling chamber. It is then extracted from the chamber during the cleaning process. This lost amount seems to be more or less the same, regardless of the amount of AgNO_3_ introduced into the reaction. Thus, if its content is too low, the silver actually taking part in the reaction can be almost zero (this most probably happened in the LEV-Ag-MS-14.87 sample, where these values are close to zero). This phenomenon was observed also in our previous work, where this decrease was only 0.5% when using three different plants [45]. This amount could be characteristic for each system. We will focus on this phenomenon in future studies when working with different plants.

The comparison of the amounts of silver before and after washing can give us very important clues about the effectiveness of stabilization, as the non-stabilized silver (being either Ag^+^ ions or Ag^0^ not effectively stabilized in the plant matrix due to lack of the plant material) is being washed out during the process. This methodology has already been described in our previous study [45].

The W value is higher than M only for the first sample containing the largest amount of plant (LEV-Ag-MS-14.87). In all other cases, the amount of silver after washing is lower than before (M > W) and the difference between M and W values increases with the introduction of larger amounts of AgNO_3_. The amount of effectively stabilized silver increased very slightly despite the fact that a large amount of it has been subjected to the mechanochemical reaction. For the sample with the highest AgNO_3_ concentration, the Ag content before washing is two times higher than after washing. However, the prolonged milling significantly improves the stabilization, as the difference between W and M value is much lower for almost the same mixture treated for 2 h (LEV-Ag-MS-1.00-120 min). The W/M ratio is plotted against lavender:AgNO_3_ ratio in Figure 4.

As can be seen, with an increasing amount of plant, the efficiency of stabilization increases. The reported value for L:A ratio (0.49) is smaller than in the case of our previous work, where three different plants were used (the values in the range 0.59–0.81 were obtained), but in that case, two hours of milling were applied. When milling for the same duration in our study (in the case of LEV-Ag-MS-1.00-120 min sample), the comparable M/W ratio of 0.84 was obtained. Considering higher L:A ratios, the values in the range 1.94–2.48 were observed for plant: AgNO_3_ ratio 10 in [45]. In our study, the lower M/W values 0.81 and 1.22 for L:A ratios 5.95 and 14.87, respectively, were observed. Most probably, the three plants applied in [45] are stronger reducing agents, thus leading to better stabilization at comparable plant: AgNO_3_ ratios.

#### 3.3.3. Scanning Electron Microscopy (SEM)

Two selected samples produced by mechanochemical synthesis were subjected to scanning electron microscopy. That is, LEV-Ag-MSW-5.95 (as in this case AgNO_3_ concentration normally used during then green synthesis was applied) and LEV-Ag-MSW-0.99 (due to highest AgNO_3_ concentration), were selected. The morphology of the two samples is compared in Figure 5.

The results obtained are a great example that the mechanochemical treatment of the same species, just at different ratios of individual components can have a determining effect on the observed morphology. At low-mag images (Figure 5a,b), fragments of the dried powder after washing of about 1 mm in size can be seen. Whereas almost homogeneous morphology with quite flat surface was obtained for LEV-Ag-MSW-5.95 sample (lavender-rich one), an inhomogeneous surface with a number of large agglomerates (traditional morphology after the mechanochemical synthesis) was observed in the case of LEV-Ag-MSW-0.99. The presence of smaller particles arranged nicely at the surface of the sample with L:A ratio 5.95 on the one hand and the presence of agglomerates in tens of microns in size on the other hand is further corroborated by the SEM images taken at higher magnification 1000× (Figure 5c,d).

In order to investigate the homogeneity of distribution of individual elements, the EDS mapping was applied. As examples, the elemental maps of silver and chlorine in selected regions for both samples are provided in the ESI (Appendix A). The different morphology reported in Appendix A also seems to influence the distribution of elements. In the sample with L:A 5.95, a homogeneous distribution of both elements can be traced down, whereas this is not completely the case in some regions of the LEV-Ag-MSW-0.99 sample (Appendix A). This difference might be caused by different amounts of AgCl in the two samples. However, the energy of the Cl-Kα peak (2.62194 keV) is very close to the energy of the Ag Lα (2.98441 keV) peak and, therefore, it is very likely that when mapping Cl distribution the signal from Ag is included in the map.

### 3.4. Comparison of Green and Mechanochemical Synthesis

#### 3.4.1. Reaction Progress

The success of the synthesis of Ag nanoparticles from AgNO_3_ should be checked from two viewpoints: (i) by the formation of Ag^0^ and (ii) by consumption of AgNO_3_. In the vast majority of the papers reporting the green synthesis of silver nanoparticles, the successful formation of Ag^0^ is proven by observing the increase of absorbance in the UV-Vis spectra in the region between 400 and 470 nm; however, the authors do not investigate whether all silver nitrate has been consumed. Usually, the termination of the synthesis is based on the measurement of UV-Vis spectra. When there is no more increase in absorbance, the reaction is considered finished (we have also applied this hypothesis in the results shown earlier). However, this might not necessarily mean that silver nitrate is exhausted, but rather that all water-soluble plant compounds with the reducing ability have been used and there are no more compounds available to reduce the residual silver nitrate. The presence of Ag^+^ ions in the prepared nanosuspension might then significantly affect the properties of the final product, namely in terms of antibacterial activity, as also Ag^+^ ion is antibacterially active [53,54]. Some authors use X-ray diffraction to prove the presence of pure Ag nanoparticles, but they use centrifugation to obtain the powder and the supernatant with possibly high content of AgNO_3_ is discarded. Of course, the XRD patterns then show just the reflection of Ag^0^. However, to determine whether or not all AgNO_3_ is consumed in the reaction, the prepared nanosuspension should be dried at low (ideally laboratory) temperature and the solid obtained should be subjected to X-ray diffraction. If there is a significant amount of non-reacted AgNO_3_, it should show up. In addition to comparing two synthetic approaches, the aim of this study is also to show the drawbacks that traditional green synthesis of Ag nanoparticles has in terms of contamination of the product with non-reacted silver nitrate.

We selected lavender plant as a reducing agent for this purpose, as there are already a number of works on the green synthesis and neither of them reports unreacted AgNO_3_ in the product [12,13,14]. In order to check whether or not there is AgNO_3_ present in the as-received reaction mixtures from the green synthesis, the obtained nanosuspensions were dried and the XRD patterns of the powders obtained are in Figure 6a. For direct comparison with mechanochemical synthesis, also the XRD patterns of as-milled powders are provided in Figure 6b. The XRD patterns are shown just for the four highest concentrations of AgNO_3_ (samples LEV-Ag-0.99 to 1.98), as the XRD patterns of plant-rich material did not provide any relevant information due to high content of amorphous plant material.

As can be seen, there is a significant amount of AgNO_3_ present in three of four investigated powders after the green synthesis (samples LEV-Ag-GS-0.99-1.48). However, the insufficient quality of the XRD patterns hampers the quantitative analysis using the Rietveld refinement for GS powders, but nevertheless, the synthesis incompletion is absolutely clear for the three samples with lowest L:A ratio. Although the diffraction peaks of AgNO_3_ were not observed for the sample with L:A ratio 1.98, neither those of Ag^0^ were not detected due to high content of plant matrix. Thus, it cannot be claimed that there is certainly no unreacted silver nitrate present in the samples with higher L:A ratios. To sum up, our hypothesis about the presence of unreacted silver nitrate after the green synthesis was found to be correct for the present experimental setup. Of course, when using other plants, the situation might be inverted and the amount of available reducing species could be satisfactory for all AgNO_3_ to be converted (e.g. no AgNO_3_ was observed after milling with *Origanum vulgare* L. [32] and *Thymus serpyllum* L. [45] for 45 minutes and 2 hours, respectively.. But nevertheless, such a proof should be provided in studies reporting the green synthesis.

The mechanochemical approach also did not result in a complete reduction of AgNO_3_ at low L:A ratios, as very intensive peaks belonging to AgNO_3_ were identified for the ratios 0.99 and 1.19 (Figure 6b). However, the content of AgNO_3_ seems to diminish when increasing the content of lavender, in accordance with [45]. In the case of L:A ratio 1.48, its peaks were barely detectable and they were absent in the mixture with the L:A ratio 1.98. The peaks corresponding to Ag^0^ can be seen in all XRD patterns, being the most intensive in the patterns of 1.48 and 1.98 ratios. Surprisingly, AgCl was not discovered in the as-received mixtures. Although this is contrary to [45], the synthesis progress was much better in that case and here AgCl diffractions might also be present, but are not visible due to an overlap with very strong ones from silver nitrate. As shown in Figure 2, AgCl was detected just after the washing process. The XRD patterns in Figure 6b (and the one of the sample with L:A ratio 1:1 treated for 2 h) were subjected to Rietveld refinement and the results with regards to phase composition can be found in Figure 7. The XRD pattern of sample LEV-Ag-MS-1.98 did not show peaks of AgNO_3_ anymore, so it was excluded from the refinement. As can be seen, prolonged milling (until two hours) brings about some improvement, as for the LEV-Ag-MS-1.00-120 sample, the content of Ag was 9.3 ± 0.6%, which is almost three times higher in comparison with 3.3 ± 1.1% detected for the LEV-AG-MS-0.99 sample. Nevertheless, the conversion still seems to be very low.

As has already been outlined in the discussion of Figure 6b, the content of Ag increased at the expense of AgNO_3_ with increasing L:A ratio. The observed trend hints to the potential exponential behavior, as confirmed also by the absence of the peaks of AgNO_3_ in the XRD pattern of LEV-Ag-MSW-1.98 sample (Figure 6b). In all analyzed samples, the calculated crystallite size for AgNO_3_ and Ag^0^ was around 70 and 25 nm, respectively and did not alter with changing L:A ratio.

It is a difficult task to directly compare the reaction progress achieved when using green and mechanochemical synthesis due to high content of amorphous plant material. In both cases, the reaction progress seems to improve with increasing L:A ratio. Interestingly, in the powders obtained by drying after green synthesis, the intensity of some crystallographic planes of unreacted AgNO_3_ become more intensive than others (they are not in line with intensities from the database) when changing L:A ratio, but this is most probably just a result of alignment during the XRD measurement.

The results obtained for green synthesis (UV-Vis spectra and XRD patterns of rough reaction mixtures) led us to the consideration which, from our point of view, might be valid for green syntheses, of generally applying a defined amount of active components extracted from bio-sources for nanoparticles synthesis. We demonstrate our ideas schematically in Figure 8.

According to Appendix A, the green synthesis velocity seems to increase with the amount of AgNO_3_ introduced. In a lavender plant matrix, many functional groups capable of reduction are available. At low concentrations of AgNO_3_, (i.e., high plant:AgNO_3_ ratio, Figure 8a), the full potential of the plant cannot be expressed, as some spots where the reduction could take place, remain unaffected. At higher AgNO_3_ concentrations, much more reducing functional groups can take part (Figure 8b). However, if the amount of introduced Ag^+^ ions is higher than the plant can reduce, the unreacted ions remain in the solution. The fact that the absorbance in the UV-Vis spectra does not increase anymore does not necessarily mean the reaction is complete because of the consumption of all silver nitrate, it can just be a result of the exhaustion of available reducing capabilities of the water-soluble plant species. This scenario is probable when using high precursor concentrations (Figure 6). Once all reducing groups are saturated, the reaction will not propagate anymore (Figure 8b).

#### 3.4.2. Grain Size Distribution

It is well known, that the individual nanoparticles are usually assembled into a larger entities, entitled grains (also terms such as, clusters or agglomerates are used). The efficient stabilization of the nanoparticles in the form of grain as small as possible is the usual strategy in the green synthetic protocols. However, during the solid-state synthesis, preventing the agglomeration phenomenon is not often the main focus, so the nanocrystals obtained are usually agglomerated into grains of micrometer size [17]. To investigate the size of grains obtained in our case using green and mechanochemical synthesis, photon cross-correlation spectroscopy (PCCS) in wet mode and laser diffraction analysis in dry mode, were applied. At first, we wanted to directly apply PCCS for both sets of samples for direct comparison, however, the powders after mechanochemical synthesis were very coarse and the results obtained were not representative. Therefore, we applied laser diffraction analysis in micro-range in the dry mode for the MS samples. The results are summarized in the ESI (Appendix A).

The size of grains is in the range of hundreds of nanometers in the case of green synthesis, whereas it is in tens of micrometers for the mechanochemical synthesis. The trend in both systems is clear: with increasing L:A ratio, the grains become smaller, thus supporting the theory about effective capping of the Ag nanoparticles obtained with larger amount of lavender species.

#### 3.4.3. Comparison of the Samples by Transmission Electron Microscopy (TEM)

The samples after both green and mechanochemical synthesis prepared with L:A ratios of 5.95 and 0.99 were analyzed by transmission electron microscopy and the resulting micrographs with selected area diffraction (SAED) patterns are shown in Figure 9. TEM results show that all samples are composed of crystalline and randomly oriented NPs embedded in organic matrix. Indexing of the diffraction rings confirms that the only crystalline phase is elemental Ag. With regards to the potential AgCl presence, very weak ring corresponding to the (200) crystallographic planes of AgCl phase can be seen in the SAED pattern taken from the LEV-Ag-GS-5.95 sample (Figure 9i), thus the presence of a small amount of AgCl can be confirmed. No AgCl has been detected in the other three samples, however it does not mean that it is not formed. In the case of the LEV-Ag-GS-0.99 sample, its absence might be caused by high amount of Ag^0^ nanoparticles and thus, potential signal of AgCl might be lost. As already explained, AgCl is mainly visible when there are small concentrations of Ag^+^ ions, due to their preferential reaction with chloride ions. The absence of AgCl in the case of MSW samples is contradictory to the XRD patterns in Figure 2, where the presence of AgCl cannot be denied. However, in the aforementioned XRD patterns, peaks of AgCl are of much lower intensity than those of Ag^0^ and the sensitivity of SAED is lower than that of XRD. For example in the study [45], the AgCl peaks were identified by XRD for two Ag:plant 1:10 samples, but were not identified by SAED. Another option is that AgCl particles are not attached to the organic matrix strongly during the dry mechanochemical synthesis and are lost during the sonication of the sample when preparing it for the TEM measurement.

The most significant difference between the samples prepared by the two approaches (GS and MS) in the present study is in the organic matrix, which is significantly thicker in the samples prepared according to the dry mechanochemical. This is most evident in the SAED pattern recorded from the LEV-Ag-MSW-0.99 sample (Figure 9d), where the thickness of the reflections of the Ag NPs are strongly damped by the thick organic matter. This is logical, as in the case of green synthesis, just the water-soluble components are applied, whereas the whole matrix is used for the mechanochemical synthesis (although part of it is washed out at the end). In both types of sample, the organic matrix acts as the reducing agent and triggers the reduction of AgNO_3_ to elemental Ag. The reduction starts at the contact between the organic matrix and AgNO_3_. The size of the initially formed Ag NPs is small, in the range of few nanometers. With the progress of Ag^0^ formation, the smaller NPs merge into larger ones. The size and morphology of the Ag NPs depends on the synthesis conditions.

In all products, bimodal size distribution is observed. In the case of the wet green synthesis, the use of lower concentration of AgNO_3_ (sample LEV-Ag-GS-5.95, Figure 9a) led to the larger difference in size between the two populations. That is, the larger Ag NPs exhibited average diameter around 60 nm, whereas the smaller ones had the diameter ≤10 nm. For the sample with the highest concentration of AgNO_3_ (sample LEV-Ag-GS-0.99, Figure 9b) the size of larger fraction of Ag NPs was smaller (average diameter around 30 nm) and the size of the smaller fractions also seemed to be somewhat smaller (<10 nm) than in the case of the LEV-Ag-GS-5.95 sample. However, the most important difference between the two samples prepared by GS was in the number of nanoparticles in each population. As can be seen from the low-magnification images in Figure 9a,b, the amount of smaller Ag nanoparticles with size below 10 nm was much higher for the LEV-Ag-GS-5.95 sample. The contribution of these small nanoparticles seems to be decisive in determining the average particle size using UV-Vis spectroscopy (Figure 1), which means that the average crystallite size of LEV-Ag-GS-5.95 was smaller than in the LEV-Ag-GS-0.99 sample. In literature, the reported size of Ag NPs usually increases with increasing Ag NPs concentration, but this is most often claimed based on the UV-Vis spectra [38,44,55]. The reports using microscopy for this purpose are rather scarce [39,56]. In both samples, a certain fraction of the finest Ag NPs (10 nm and below) remains immobilized inside the organic matrix. In the wet synthetic approach, the Ag NPs develop octahedral morphologies with rounded edges. The presence of {111} twin boundaries is observed in many particles, which may be attributed to the presence of chlorine [57,58,59]. In the samples synthesized under wet conditions, the distribution of Ag NPs seems relatively homogenous within the organic matrix.

The mechanism of Ag NPs formation is similar in the samples prepared using the dry mechanochemical synthesis, however, the mobility of Ag is more restricted in this case. Areas with both higher and lower local concentration of Ag NPs were observed in TEM. Also, here the presence of very small Ag NPs is confirmed. The visibility of these smallest nanoparticles is worse in micrographs taken at lower magnifications due to higher thickness of the organic matrix. The presence of crystalline Ag NPs with sizes around and below 10 nm is observed in images taken at higher magnifications and in thin parts of the samples. In mechanochemically produced samples, the morphology of the larger Ag NPs is more irregular or spherical, which may be attributed to coarsening under dry conditions. The difference in Ag NPs size is not that obvious in mechanochemically synthesized samples with different L:A ratio, however, the presence of a higher amount of the finest Ag NPs (<5 nm) was observed in the samples when a smaller amount of AgNO_3_ was introduced (sample LEV-Ag-MSW-5.95), thus being in accordance with the results from green synthesis and those reported for other plants in [45].

#### 3.4.4. Comparison of Antibacterial Activity

The use of elemental silver as antibacterial agent has its roots in ancient time [60]. Its activity is influenced by various factors, namely concentration of nanoparticles and capping agents, shape and size of NPs [61,62,63,64]. The exact mechanism of antibacterial action of Ag NPs is still a subject of scientific discussion, however, three mechanisms are usually considered: (a) irreversible damage of bacterial cell membrane through direct contact; (b) generation of reactive oxygen species (ROS); and (c) interaction with DNA and proteins [65,66,67,68,69]. The reaction of Ag NPs with sulphur and phosphorus-containing biomolecules seems to be the main reason of their antibacterial properties [70,71]. In the present work, we have prepared Ag NPs using the lavender plant. The antibacterial activity is reported in neither of the papers [11,12,13,14] using the same plant for the Ag NPs. The RIZD values obtained for the samples LEV-Ag-GS-0.99 and 5.95 are in accordance with those reported in [47], where the values above 90% for concentration 5 mM AgNO_3_ and above 80% for the concentration 2.2 mM AgNO_3_ were reported. The antibacterial activity of lavender essential oil is widely known and is also a focus of recent research [72,73,74].

The antibacterial activity of LEV-Ag samples prepared by green and mechanochemical synthesis was evaluated against two bacterial strains, namely Gram-positive *Staphylococcus aureus* and Gram-negative *Escherichia coli*. The photographs of the Petri dishes after the incubation with the obtained LEV-Ag samples is shown in the Appendix A. The results expressed as relative inhibition zone diameter (RIZD) are presented in Figure 10.

The antibacterial activity of the samples prepared using green synthesis was higher in all cases. The difference was not so significant at high L:A ratios, but became really large as the concentration of AgNO_3_ in the mixtures increased. This huge difference at low L:A ratios can be now ascribed to the presence of unreacted AgNO_3_ in the samples after green synthesis. The highest L:A ratio when the results were more-or-less comparable was 2.97. The antibacterial activity did not increase further in the mechanochemically prepared samples, in accordance with very slight increase in the amount of silver after washing in LEV-Ag-MSW samples (Table 2). The use of the highest concentration of AgNO_3_ leads to the improvement in the antibacterial activity of the MSW sample against *E. coli*, but this did not happen in the case of *S. aureus*. The activity was only slightly improved when longer milling for two hours was applied, despite the fact that the actual amount of stabilized silver significantly increased (Table 2). On the other hand, the activity of the samples after green synthesis increased further almost linearly. A calculation using the RIZD values obtained for MSW samples and the Ag content determined by AAS yielded the hypothetical values (marked by dashed lines in Figure 10), which could be obtained if all Ag subjected to mechanochemical reaction is effectively stabilized (to do this, it was hypothesized that the antibacterial activity is linearly proportional to the content of Ag in the sample). Surprisingly, the RIZD values obtained for green synthesis were almost completely in line with the calculated values for *E. coli* and were slightly lower, but still close, to those detected for *S. aureus*. In general, the RIZD values of both in a recent study and [47] were slightly lower for *E.coli*.

As mentioned earlier, the antibacterial activity is influenced not only by the concentration, but also by the size of the nanoparticles. However, there seems to be no relationship between the crystallite size of Ag NPs in the MSW samples (reported in Figure 3) and the antibacterial activity in our study (this can be mainly seen from the comparison of the results obtained for the samples with low L:A ratio). Whereas the crystallite size dramatically decreased until L:A ratio 2, the antibacterial activity for MSW samples stays more or less at the same level.

Although the results achieved for mechanochemically prepared materials are worse, it has to be noted that the actual content of AgNO_3_ in the green-synthesized samples is not known (the results for which the presence of AgNO_3_ was proven in LEV-Ag-GS samples is encircled in blue in Figure 10). In the case of mechanochemical synthesis, we proved a significant content of AgNO_3_ after the milling and then washed it out, hence we can definitely say that the observed antibacterial activity is due to the silver nanoparticles produced. Such a statement cannot be made for the Ag NPs prepared by green synthesis due to the unknown contribution of non-reacted AgNO_3_. In general, the antibacterial activity was slightly better against *S. aureus* than for *E. coli*, which is in accordance with our previous work [45]. However, a contrary situation has been reported in ref. [14,32,46].

Further considerations taking into account calculations of the actual amount of silver in the well-used during antibacterial tests and the discussion of the observed antibacterial activity with regards to this parameter can be found in the ESI (see Appendix A and related discussion).

In general, the bio-mechanochemical synthesis of Ag NPs was not completed in 15 min (and neither in 120 min, Figure 6) when using *Lavandula angustofilia* L. plant, thus most probably leading to a lower amount of effectively stabilized silver in comparison with other plants (e.g., *Origanum vulgare* L. or *Thymus serplyllum* L. at the same Ag:plant ratios), where the conversion was almost complete (no AgNO_3_ was evidenced in the XRD patterns). This could lead to improved antibacterial activity, although in this study we have shown that if the AgNO_3_ concentration is increased beyond a certain L:A ratio, further improvement in the antibacterial activity is not achieved. However, this is surely connected with the reductive ability of the plant. Thus, in the future we are planning to prepare a composite with the largest amount of Ag incorporated possible by using a strong reductive plant (e.g., *Thymus serplyllum* L. or *Origanum vulgare* L.) and comparing the activity of the mechanochemically synthesized products with that of Ag NPs prepared by the green synthesis using the same plant. Mechanochemistry surely has the potential to outperform the green synthesis, maybe it also does in the present case, but we do not have the information about AgNO_3_ content in the LEV-Ag-GS samples. Moreover, the wet stirred media milling applied in the second stage (after ball milling in a planetary ball mill—an approach that is used when preparing nanoparticles for biomedical applications [75,76]) could lead to a better effect, as was demonstrated for stabilization of Ag NPs in the past [31].

## 4. Conclusions

Green synthesis of silver nanoparticles has been tackled from almost any perspective possible in the last years. However, a direct comparative study comparing one-step-solid-state bio-mechanochemical synthesis with the traditional plant-extract green synthesis has been missing until now. Herewith, we have successfully prepared Ag nanoparticles using both approaches by applying the common *Lavandula angustofolia* L. (lavender) plant as a reducing agent. Seven different lavender:AgNO_3_ mass ratios were used to investigate the performance of both methodologies at low and at high precursor concentrations. The green synthesis was very quick (in the slowest case, it lasted just 8 min), and its velocity increased with increasing AgNO_3_ concentration. By drying the crude reaction mixtures, we succeeded in showing the presence of unreacted silver nitrate, namely when higher concentrations were applied. Such an investigation is missing in the majority of papers reporting the green synthesis and often its presence changes the final performance of the Ag NPs produced. The un-reacted AgNO_3_ was detected also in the case of mechanochemical synthesis lasting for 15 min, however, it was effectively removed by washing with distilled water. The nanocrystalline character of elemental silver was confirmed both by Rietveld refinement of X-ray diffraction data and TEM analysis, with the particles showing bimodal size distribution (the larger in tens of nm and the smaller below 10 nm in size). In addition to Ag nanoparticles, a small amount of AgCl was identified in the mechanochemically synthesized samples. With increasing lavender content, the amount of silver stabilized in the solid phase of the mechanochemically prepared powders increased, however when the amount of plant was too high, the content of AgCl was relatively higher in comparison with the amount of Ag^0^. We are of the opinion that limited amount of AgCl is formed both when using mechanochemical or green synthesis and its amount is limited by the content of chlorine in lavender plant.

Finally, all the prepared products exhibited antibacterial activity against both *E. coli* and *S. aureus*. In general, slightly better antibacterial activity was evidenced for the Ag NPs prepared by green synthesis and the activity increased with increasing precursor concentration. However, the unreacted silver nitrate might have been the reason for the better performance of Ag NPs prepared using the solution-based method. In general, it seems that upon using a properly optimized experimental setup (with regards to both type of reducing agent and milling conditions), there is still a space for the mechanochemical synthesis to outperform the green synthesis in terms of reaching better antibacterial activity and better stability due to the solid character of the products.

## Figures and Tables

**Figure 1 nanomaterials-11-01139-f001:**
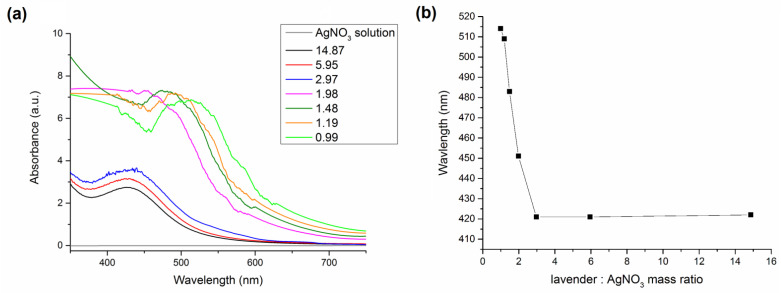
(**a**) Ultraviolet-visible (UV-Vis) spectra of the final products of green synthesis at various lavender:AgNO_3_ ratios; (**b**) dependence on the wavelength of absorbance maximum on the lavender:AgNO_3_ ratios.

**Figure 2 nanomaterials-11-01139-f002:**
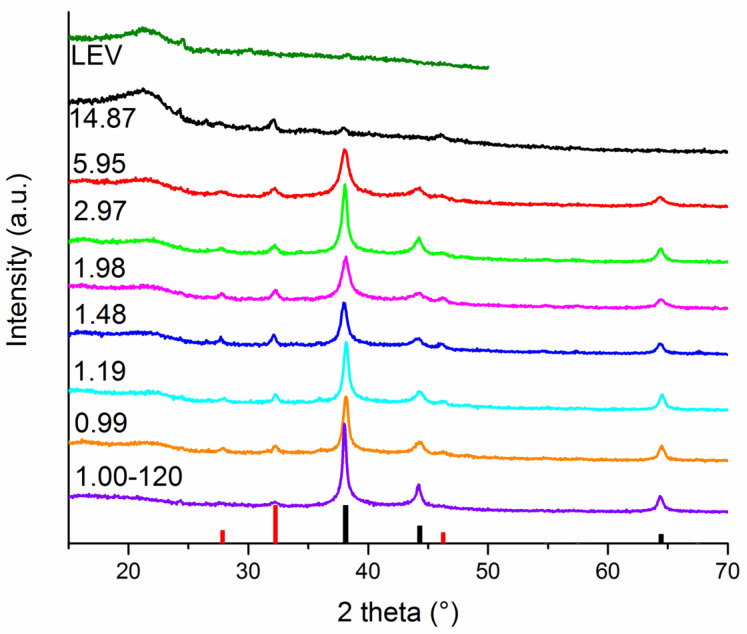
X-ray diffraction (XRD) patterns of all LEV-Ag-MSW (washed samples after mechanochemical synthesis) milled for 15 min after washing. The numbers on the left of each XRD pattern correspond to lavender:AgNO_3_ (L:A) mass ratio. The XRD pattern of pure lavender plant (*Lavandula angustifolia* L., LEV) is also provided. At the bottom, red bars correspond to cubic AgCl (ICDD 71-5209) and black to cubic Ag (ICDD 65-2871).

**Figure 3 nanomaterials-11-01139-f003:**
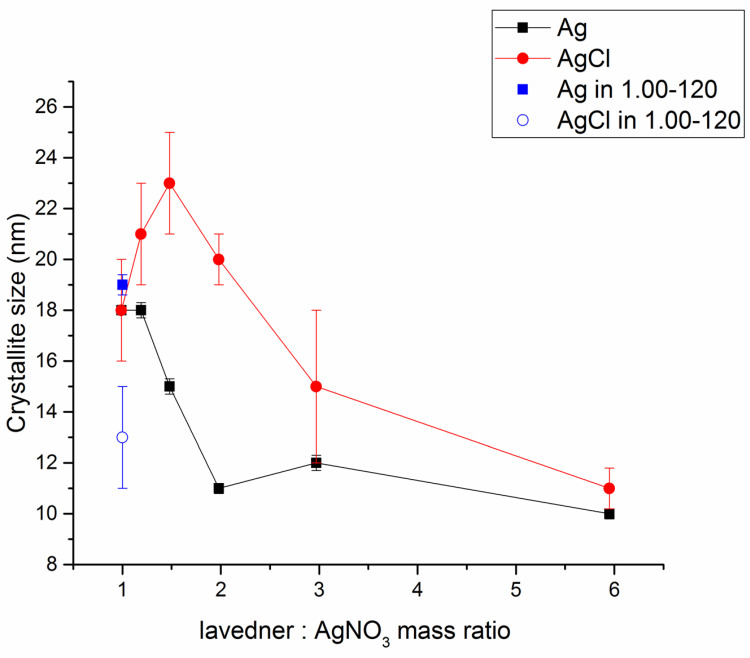
Changes in crystallite size of Ag and AgCl with the lavender:AgNO_3_ ratio in all LEV-Ag-MSW samples.

**Figure 4 nanomaterials-11-01139-f004:**
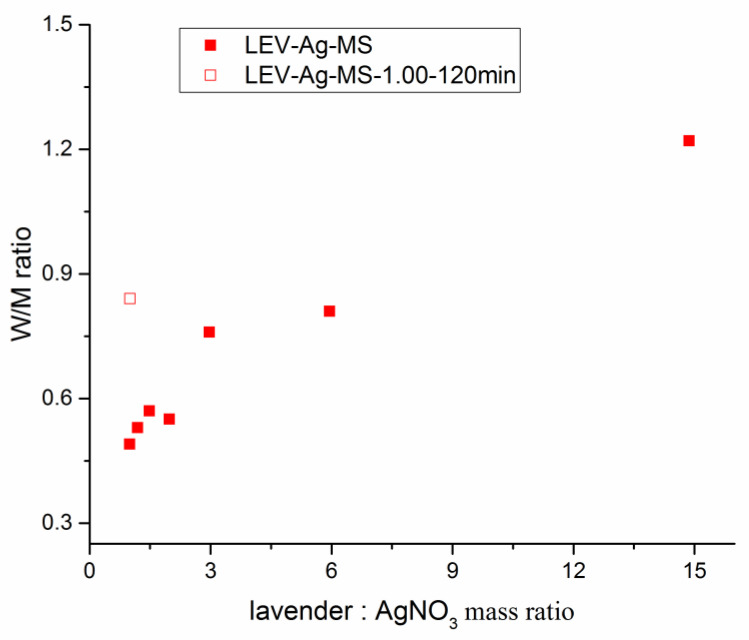
The dependence of W/M ratio determined from atomic absorption spectrometry on lavender:AgNO_3_ mass ratio used in the mechanochemical synthesis.

**Figure 5 nanomaterials-11-01139-f005:**
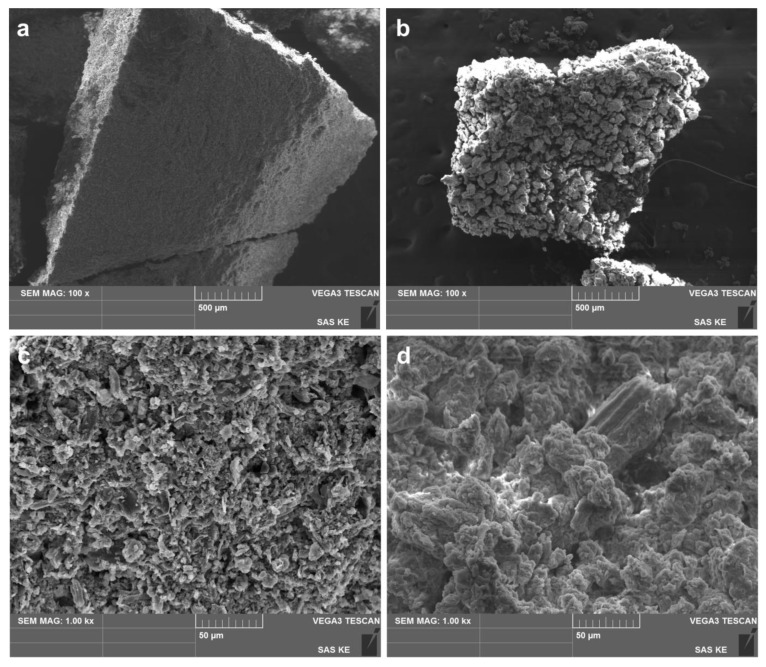
Scanning electron micrographs of LEV-Ag-MSW-5.95 (**a**,**c**) and 0.99 (**b**,**d**) samples taken at (**a**,**b**) lower, (**c**,**d**) higher magnification.

**Figure 6 nanomaterials-11-01139-f006:**
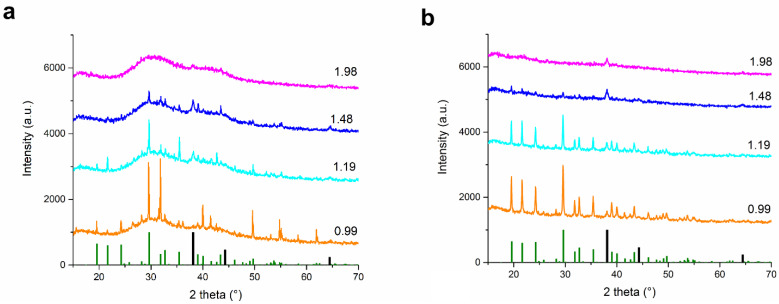
XRD patterns of the dried reaction mixtures after the green synthesis (**a**) and as-milled powders after the mechanochemical synthesis (**b**) using the four highest lavender:AgNO_3_ mass ratio. At the bottom, green bars correspond to orthorhombic AgNO_3_ (ICDD 74-4790) and black to cubic Ag (ICDD 65-2871).

**Figure 7 nanomaterials-11-01139-f007:**
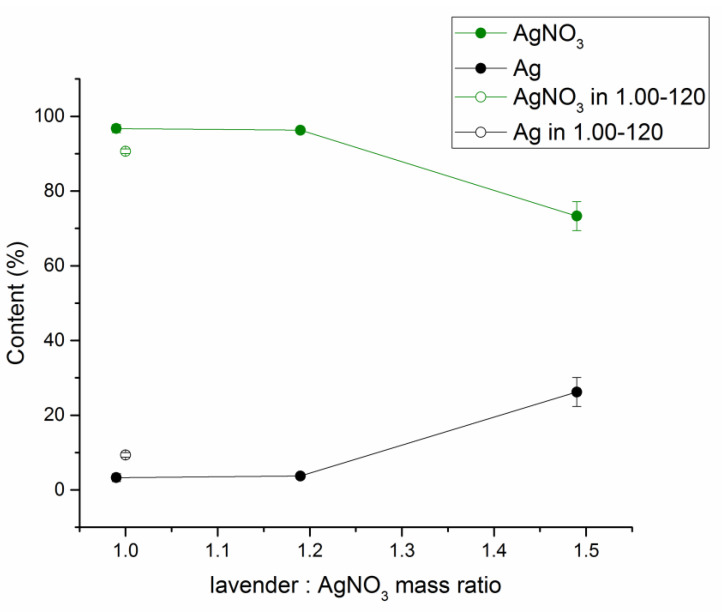
The content of Ag and AgNO_3_ in the mechanochemically prepared samples with lowest lavender:AgNO_3_ ratios.

**Figure 8 nanomaterials-11-01139-f008:**
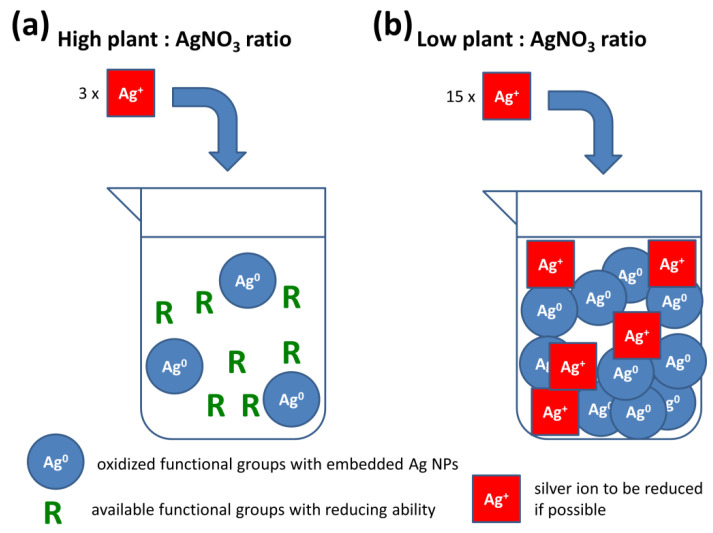
Schematic illustration of the events taking place in the flask during the green synthesis of nanoparticles at (**a**) high and (**b**) low plant:AgNO_3_ ratios.

**Figure 9 nanomaterials-11-01139-f009:**
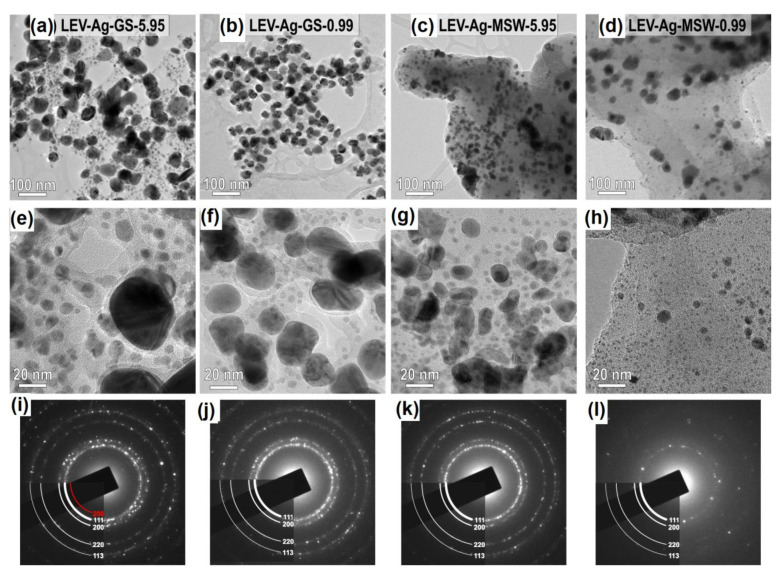
Transmission electron microscopy (TEM) images of Ag NPs prepared by green or mechanochemical synthesis at (**a–d**) lower and (**d–h**) higher magnification; (**i–l**) SAED patterns shown in the bottom row reveal that in all samples, the main reflections stem from elemental Ag, only in the sample (**a**) LEV-Ag-GS-5.95, a weak ring from (200) planes of the AgCl phase (marked with red color) can be observed.. The results for the corresponding samples are presented as follows: (**a**,**e**,**i**) LEV-Ag-GS-5.95; (**b**,**f**,**j**) LEV-Ag-GS-0.99; (**c**,**g**,**k**) LEV-Ag-MSW-5.95; (**d**,**h**,**l**) LEV-Ag-MSW-0.99. Sample marks are also given inthe figures (**a–d**).

**Figure 10 nanomaterials-11-01139-f010:**
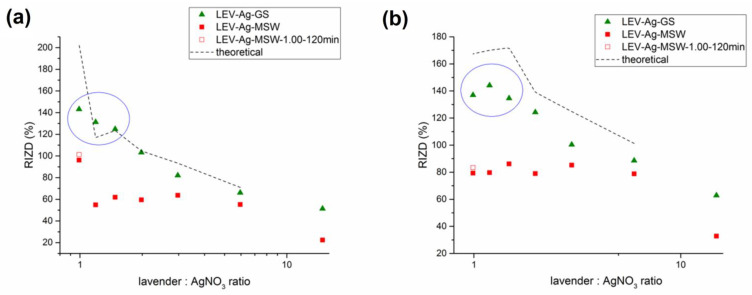
Antibacterial activity of green-synthesized (GS) and washed mechanochemically synthesized (MSW) Ag nanoparticles using lavender. The activity is presented as relative inhibition zone diameter (RIZD) against bacteria (**a**) *E. coli* and (**b**) *S. aureus* as a function of lavender:AgNO_3_ ratio. The dash line describes the hypothetical antibacterial activity if all silver introduced into the reaction would be stabilized (based on the Ag contents determined from atomic absorption spectrometry (AAS). Blue circles show the regions, where the silver nitrate certainly contributes to the antibacterial action of the GS samples.

**Table 1 nanomaterials-11-01139-t001:** Masses of lavender and AgNO_3_ in both green (GS) and mechanochemical syntheses (MS) and corresponding lavender:AgNO_3_ (L:A) mass ratios.

	Green Synthesis			Mechanochemical Synthesis		Lavender: AgNO_3_ (L:A) Mass Ratio
Sample	Lavender Mass (mg)	AgNO_3_ Mass (mg)	AgNO_3_ Concentration (mM)	Lavender Mass (g)	AgNO_3_ Mass (g)	
LEV-Ag-14.87	15	1.009	2.2	2.8109	0.189	14.87
LEV-Ag-5.95	15	2.523	5.5	2.5681	0.4318	5.95
LEV-Ag-2.97	15	5.045	11	2.2449	0.755	2.97
LEV-Ag-1.98	15	7.568	16.5	1.9939	1.006	1.98
LEV-Ag-1.48	15	10.090	22	1.7935	1.2064	1.48
LEV-Ag-1.19	15	12.613	27.5	1.6296	1.3703	1.19
LEV-Ag-0.99	15	15.135	33	1.4932	1.5067	0.99
LEV-Ag-1.00-120	-	-	-	1.5000	1.5000	1

**Table 2 nanomaterials-11-01139-t002:** Mass losses during washing, Ag content in the milled (M) and washed (W) samples determined by atomic absorption spectrometry and the corresponding W/M ratio.

Sample	Mass Loss during Washing (%)	Theoretical (Initial) Ag Content in the Sample (%)	Ag Content in the Milled Sample (M) (%)	Ag Content in the Milled Sample after Washing (W)(%)	W/M Ratio
LEV-Ag-14.87	n/a	4	0.09	0.11	1.22
LEV-Ag-5.95	69.4	9.1366	8.73	7.11	0.81
LEV-Ag-2.97	59.3	15.9666	14.31	10.9	0.76
LEV-Ag-1.98	54.6	21.2933	21.75	12.1	0.55
LEV-Ag-1.48	51.3	25.5266	22.1	12.8	0.57
LEV-Ag-1.19	46.1	29.0033	25.5	13.6	0.53
LEV-Ag-0.99	41.8	31.8733	30.6	15.1	0.49
LEV-Ag-1.00-120 min	n/a	31.89	27.4	23.3	0.84

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
