# Peer review of "Mechanochemistry as an Alternative Method of Green Synthesis of Silver Nanoparticles with Antibacterial Activity: A Comparative Study"

_nanomaterials, 2021, doi:10.3390/nano11051139_

Round 1

Reviewer 1 Report

This work describes the advantages and disadvantages of two methods: mechano- chemistry and green synthesis of Ag nanoparticles using plant extracts  in relation to the concentration of AgNO3 precursor on several samples. TThe combination of mechanochemistry and green 
synthesis could be advantageous from environmental point of view  by the  use of natural material as reducing agent and by offering the the possibility to perform the synthesis in the solid-state in one step (without the need of a separate step of extract preparation). Antibacterial activity was demonstrated.

The paper is well constructed, and the results sustain the conclusions. There are only minor comments related to the need to include a discussion on the mechanism of materials bacteria interaction.

The paper can be published after this correction.

Author Response

We thank the reviewer for the positive evaluation of our work. We have added a short paragraph to describe the interaction between the prepared material and studied bacteria and added relevant references reporting interesting facts on this topic.

Reviewer 2 Report

The paper “Mechanochemistry as an alternative method to green synthesis of silver nanoparticles with antibacterial activity: A comparative study” reports the comparative study of the synthesis of silver particles by a green synthesis and a mechanochemical synthesis using lavender as reducing agent for silver ions.

In my opinion, while the formation of nanoparticles is sufficiently demonstrated for the green synthesis, by showing UV-Vis spectra, this is not convincing for the mechanochemical synthesis. In fact, UV-Vis spectra could not be recorded and the SEM images only show large aggregates at magnification not larger than 1000. Only TEM images show nanoparticles. This is probably because, as mentioned in the experimental section, the powder sample was sonicated in ethanol before the TEM analysis, thus disrupting the large aggregates.

Given the above consideration, the paper cannot be accepted in the present form, unless major revisions of the manuscript occur.

Other points that need attention are:

  • In section 2.2.2 the authors should clarify in what form AgNO3 is introduced in the mixture, whether in powder, solution or other
  • In section 3.2.1 the authors claim that the synthesis is faster when the Ag concentration is higher. While this can be implied from fig. S1-d from the different slope of the time curves, kinetics considerations in this case should report actual absorbance values, making sure to work in the range of linearity of the Lambert-Beer law (A<1). As it stands, it can only be claimed that the reaction goes to completion faster.
  • Further in the same section, the authors claim that the red-shift in the plasmon resonance position is due to aggregation because this is a concentration related phenomenon. I think there is no evidence given on that (i.e. SEM images) so it could just be that particles grow bigger.
  • In section 3.3.1 the authors interpret broad and low-intensity Ag peaks at XRD by having very small Ag nanoparticles. This statement should be more supported.
  • In the same section the authors discuss the role of AgCl in the mechanochemical synthesis. As Cl ions come from lavender, the authors should comment on why the same role is not played by AgCl in the green synthesis.
  • In Table 2, does the theoretical amount stand for the initial silver amount?
  • In the same table, Ag content M is higher than theoretical one (21.75 vs 21.2933) for sample LEV-Ag-1.98. How can that be?
  • In the same table, W/M ratio is higher than 1 for sample LEV-Ag-14.87, as also highlighted by the authors in the text. How can that be?
  •  

Author Response

We thank the reviewer for a detailed evaluation of our work. Please find the attached Word document with all the answers.

Reviewer 3 Report

The work reports the results on the production of Ag by reduction of AgNO3 with plant extracts (in solution) or Lavander plant powders (in solid-state). 

We are in the field of green chemistry, and this is certainly current and interesting.

Two methods have been analyzed and reported by the authors: the first uses plant extracts (green chemistry) in a solution synthesis. The second, proposed as a novelty by the authors, uses the fine trituration of AgN03 in the presence of the Lavander plant (bio-mechanical method), which is dry (reduction reaction in the solid-state).

The results indicate that the bio-mechanical method can produce Ag from AgNO3, although not entirely satisfactory.

An interesting result could be that the authors report data on residual AgNO3 in the solution preparations by green-chemistry. This aspect has been little studied so far.

Characterizations by microscopy confirm the not-homogeneity of the materials and the formation of the Ag.

In my opinion, the paper should be deeply revised in the following aspects before publication:

- The work is too long and very difficult to read for a non-expert in mechanochemical chemistry. There are too many experimental details and conjectural considerations in the discussions.

- All the procedures and the changes of concentrations and others parameters had to be better schematized, arriving at the results. Experimental details should be reported in the experimental part.

- Why do the authors discuss the antibacterial activity of these raw materials? Is there a particular use of these activities? ? If so, it must be addressed with pertinent comparisons.

- English needs to be revised.

Reviewer 4 Report

The manuscript presents a study of mechanochemistry method to synthesize Ag nanoparticles with antibacterial activity, as compared to traditional green synthesis. The overall impression it gives is a little confuse, as some information is given rather qualitatively and several references are made to other works without the detailed data (for instance, authors mention but do not give the SPR for electronically-coupled Ag particles, as compared to individual nanoparticles, in the paragraph below Figure 1).   In particular, table 1 needs a more clear differentiation between the columns related to green synthesis, to mechanochemical synthesis and to both. Also, it deserves to be in the "2.2. Synthesis of Ag NPs" section. Even, reference samples such as pure lavender and LEV:Ag = 1 milled 120 minutes could be included in this technology table and in the results figures (e.g. in Fig. 7). Additionally, please revise writing the "2.4. Actibacterial activity" section, as some repetition seems to be present at the end of the first paragraph and compatibility of temperatures and times is not clear. Perhaps a numbering (or bullet) listing could clarify the actibacterial-properties-evaluation steps.   Figure 1 (a) shows absorbance spectra of GS samples, showing a SPR-band evolution according to the L:A ratio. Description of this figure mentions intensive peaks located in the 413-474 nm range. However, large maxima are visible at higher wavelengths (see for example the 0.99 curve, which has a maximum at around 510-520 nm). What are they due to?   In general, a qualitative change is visible between L:A=1.98 and 2.97, suggesting that these ratios are critical. For instance, the comment of Fig.3 (a) mentions a general decrease of Ag content with the increasing amount of plant, with an exception in the 2.97 ratio. But exceptions seem rather to be the 1.48 and 1.98 ratios, which show exceptionally low Ag contents. Similarly, the exception in Fig. 3 (b) might be 1.98 rather than the 2.97 ratio named in the body. Could authors argue against the one or the other proposal? For this, a measurement of the grain size before washing could be useful. Could authors propose any reason for this kind of behavior?

Round 2

Reviewer 2 Report

I am satified with the authors remarks and the improvements they made on the paper. I recommend publications with minor revisions as follows:

line 260: AgNO3 should be AgNO3

line 381: subic should read cubic

Author Response

We thank the reviewer for positive evaluation of the performed modifications and careful eyes. The two mentioned mistakes have been corrected.

Reviewer 3 Report

The paper can be published after these modifications

Author Response

Thank you for positive evaluation.